# Global, regional, and national burden of nonalcoholic fatty liver disease among adults aged ≥ 45 years: A comprehensive analysis of epidemiological trends and projections to 2035

Qian Wang[1‡], Jieru Guo[2‡], Shuang Liu[3‡], Xuebin Cao[4], Zhirong Guo[4], Long Rui[3], Liu Zheng[1], Chenyang Wang [ID][4]*

**1** Department of Clinical Laboratory, The Hospital of 82nd Group Army PLA, Baoding, Hebei, China, **2** 926 Hospital of Joint Logistics Support Force, PLA, Kaiyuan, Yunnan, China, **3** Department of Gastroenterology, The Hospital of 82nd Group Army PLA, Baoding, Hebei, China, **4** Central Laboratory, Department of Cardiology, The Hospital of 82nd Group Army PLA, Baoding, Hebei, China

‡ These authors contributed equally to this work and share first authorship.
* wangbio_2004@163.com

## Abstract

### Background

Nonalcoholic fatty liver disease (NAFLD) has emerged as the leading cause for chronic liver diseases around the globe, disproportionately affecting aging populations. This research focused on the global burden of NAFLD in adults aged 45 and older from 1990 to 2021, with projections extending to 2035.

### Methods

Using data from the Global Burden of Disease (GBD) Study between 1990 and 2021, we assessed the incidence, prevalence, mortality and disability-adjusted life years (DALYs) related to NAFLD in adults aged 45 and older in 204 countries and territories. To evaluate the underlying drivers including demographics and lifestyle, Bayesian age-period-cohort (BAPC) modeling was employed.

### Results

In 2021, the worldwide prevalence of NAFLD has reached 48.35 million cases (with a 95% uncertainty interval of 44.23 to 52.36 million). Among individuals aged ≥ 45 years, age-standardized incidence rose by 18.3% (EAPC = 0.53) from 1990 to 2021, while prevalence increased by 24.5% (EAPC = 0.74). Mortality and DALYs also climbed, with Egypt, Mongolia, and Andean Latin America bearing the highest burdens. A bell-shaped Socio-Demographic Index (SDI) correlation emerged, peaking in medium-SDI regions (e.g., North Africa, Middle East). Projections indicate persistent

**Data availability statement:** The original data of this study are all from the publicly available GBD database (URL: https://vizhub.healthdata.org/gbd-results/), and do not include personal information such as patients' names or IDs. The population data was downloaded from WPP (https://population.un.org/wpp/). Therefore, this study does not require an additional ethical statement. In addition, the code in this study has been uploaded to Github (URL: https://github.com/shuangliu2025/R_FOR_GBD_ANALYSIS/tree/main).

**Funding:** These findings are the result of work supported by "Theater Army 2023 Medical Autonomous Research Project" grant from Support Department of the Central Theater Army. Project Number: 2023LC09. China. The full name of the authors who received this award are Chenyang Wang. The views expressed in this paper are those of the authors, and no official endorsement by the Central Theater Army PLA is intended or should be inferred. The funders had no role in study design, data collection and analysis, decision to publish, or preparation of the manuscript.

**Competing interests:** The authors declare no conflicts of interest in preparing this article.

female predominance, with ASIR expected to rise to 826.11 (women) vs. 665.72 (men) per 100,000 by 2035.

## Conclusions

This analysis explored the global burden of NAFLD in people aged 45 years and older from 1990 to 2021, demonstrating significant epidemiological changes. Age-standardized incidence and prevalence rates rose by 18.3% and 24.5%, respectively, with the most pronounced burden observed in middle-to-high SDI regions attributable to aging populations. Although women exhibited higher incidence rates, mortality rates remained consistently elevated among men, underscoring unmet intervention needs. Projections to 2035 indicate increasing incidence (particularly in women) alongside moderate declines in mortality and DALYs, underlining the requirement for prevention strategies that are specific to age and gender.

## Introduction

Non-alcoholic fatty liver disease (NAFLD) has undergone a dramatic epidemiological transition, currently the most common chronic liver disease globally, affecting about 25–30% of adults around the world [1]. Its pathological spectrum ranges from simple hepatic steatosis to non-alcoholic steatohepatitis (NASH), fibrosis, cirrhosis, and hepatocellular carcinoma (HCC) [2]. Recent evidence indicates that NAFLD accounts for approximately 2.4% of global mortality from liver diseases, surpassing both viral hepatitis and alcoholic liver disease in many developed nations [3].

The pathophysiological intersection between NAFLD and aging presents unique clinical challenges due to age-related alterations in hepatic lipid metabolism [4], characterized by diminished β-oxidation capacity, mitochondrial dysfunction, and increased pro-inflammatory cytokine production with aging [5]. Clinical studies have demonstrated that fibrosis progression rates accelerate by 4–5% per decade after age 40, culminating in a 15-fold elevated risk of HCC by age 65 [6]. Furthermore, aging populations with NAFLD frequently exhibit metabolic multimorbidity, with over 70% of patients ≥ 45 years presenting concurrent type 2 diabetes, hypertension, or cardiovascular disease [7].

Despite these critical interactions, comprehensive assessments of NAFLD burden specifically among middle-aged and older adults remain scarce. Existing Global Burden of Disease (GBD) analyses largely consolidate NAFLD estimates across wide age brackets, potentially obscuring important epidemiological variations within high-risk subgroups [8]. This knowledge gap persists even as demographic aging reshapes global health priorities-the population aged ≥ 60 years is projected to double by 2050 and is expected to constitute the majority requiring liver-related healthcare [9].

Given the accelerated disease progression observed in middle-aged and older adults, combined with the growing global prevalence of metabolic risk factors, there is an urgent need for comprehensive age-stratified analyses of NAFLD burden. While

previous Global Burden of Disease studies have provided valuable insights into overall NAFLD epidemiology, they have not specifically examined the unique patterns and drivers affecting adults ≥45 years-a population experiencing the most rapid demographic growth globally. This knowledge gap limits our ability to develop targeted prevention strategies and allocate healthcare resources effectively for this high-risk demographic. Therefore, this study aims to provide the first comprehensive assessment of NAFLD burden specifically among adults aged ≥45 years, utilizing the most recent Global Burden of Disease 2021 data to inform evidence-based policy development.

## Methods

### Data sources

We made use of information from the Global Burden of Diseases and Injuries Study (GBD, URL: https://vizhub.healthdata.org/gbd-results/) conducted in 2021, which synthesizes epidemiological information from 204 nations and territories [10]. The data obtained from the GBD database did not require informed patient consent and was publicly available.

GBD 2021 harmonized NAFLD case definitions by integrating country-specific diagnostic modalities. For 87 countries with biopsy/imaging studies, cases required histologic steatosis (≥5% hepatocytes) or imaging-confirmed hepatic fat fraction >5% by MRI-PDFF or ultrasound. In remaining nations, FLI ≥ 60 was applied as a surrogate, validated against local imaging cohorts where available (e.g., FLI sensitivity/specificity = 0.73/0.86 in European and 0.68/0.81 in Asian populations) [11]. GBD's DisMod-MR 2.1 tool adjusted for cross-country diagnostic heterogeneity by incorporating covariates such as healthcare access and obesity prevalence [12]. Mortality estimates incorporated vital registration systems, verbal autopsy data, and cancer registry records coded to ICD-10 codes K75.8 and K76.0.

Inclusion criteria required: (1) age ≥ 45 years at diagnosis; (2) NAFLD defined per FLI ≥ 60 or imaging-confirmed hepatic steatosis (≥5% hepatocyte involvement); and (3) residency in a GBD-listed country/territory. Exclusion criteria, applied through GBD's hierarchical cause-of-death modeling, included: (1) secondary hepatic steatosis due to alcohol consumption > 20g/day (men) or>10g/day (women); (2) viral hepatitis B or C coinfection; (3) drug-induced steatosis (corticosteroids, methotrexate, amiodarone); (4) hereditary metabolic disorders (Wilson disease, alpha-1 antitrypsin deficiency); and (5) other chronic liver diseases taking precedence in GBD's mutually exclusive disease hierarchy.

This study focuses on adults aged ≥45 years based on clinical and public health considerations. Beginning in mid-life, metabolic alterations—such as increased insulin resistance, hormonal changes, visceral adiposity, and sarcopenia—promote hepatic lipid accumulation and elevate NAFLD risk. After age 45, fibrosis progression accelerates, with each decade increasing fibrosis risk by 4–5%, and cirrhosis and HCC incidence rise substantially. This group also exhibits high multimorbidity; over 70% of NAFLD patients have concurrent metabolic conditions, compounding mortality risk. Globally, aging populations make this age group a major driver of NAFLD-related healthcare burden [13]. Prior studies often overlook age-specific patterns, limiting targeted interventions. Focusing on this cohort allows clearer insight into demographic and epidemiologic drivers and supports cost-effective early detection and long-term policy planning.

### Statistical analysis

Age-standardized rates were calculated using the global standard population set by the World Health Organization [14]. Uncertainty intervals (UIs) were estimated by accounting for sampling error, diagnostic variability, and model uncertainty through 1000 draws from the Bayesian posterior distribution [13].Projection modeling was generated using Bayesian age-period-cohort (BAPC) models, which incorporated demographic changes from UN World Population Prospects 2022, Healthcare access metrics (Universal Health Coverage index) [15]. The Bayesian Age-Period-Cohort (BAPC) model produces more reliable predictions of global disease burden trends by leveraging the similarity of age, period, and cohort effects across adjacent time intervals. It applies a second-order random walk prior to smooth these three types of effects and derives posterior rate estimates through Bayesian inference. The model uses integrated nested Laplace

approximation (INLA) to estimate marginal posterior distributions, which mitigates mixing and convergence issues often associated with traditional Markov chain Monte Carlo sampling in Bayesian analysis [16]. To ensure smoothness, the BAPC model assigns independent mean-zero normal distributions as priors to the second-order differences of all effects, with the prior distribution for the age effect specified as follows:

$$\mathbf{f}(\alpha \mid \mathbf{k}_\alpha) \propto \mathbf{k}_\alpha^{\frac{t-2}{2}} \exp\left\{-\frac{\mathbf{k}_\alpha}{2} \sum_{i=3}^{1} [(\alpha_i - \alpha_{i-1}) - (\alpha_{i-i} - \alpha_{i-2})]^2\right\}$$

Second-order random walk (RW2) priors were assigned to age, period, and cohort effects with precision hyperparameters following Gamma(1, 0.00005) distributions. Sum-to-zero constraints were implemented to resolve identifiability issues inherent in age-period-cohort models. Bayesian inference utilized Integrated Nested Laplace Approximation (INLA) for computational efficiency. Model selection employed the Deviance Information Criterion (DIC), with final models achieving DIC values <15,000 across all regions. Convergence was assessed using effective sample size (ESS > 1000) and Gelman-Rubin potential scale reduction factors (<1.1). Model validation involved comparing predicted versus observed rates from 1990–2021, achieving mean absolute percentage errors <5% across 95% of country-years.

Geographic differences were analyzed based on Socio-demographic Index (SDI) groups [17], regions defined by the WHO (like WPRO, SEARO, EURO, etc.), and estimates at the country level. Measures of health inequality, such as the Slope Index of Inequality (SII) and Concentration Index (CI), were calculated. All analyses and result visualization were conducted using the software R (version 4.3.3).[18].

## Results

### Global burden of NAFLD in 2021

According to the 2021 Global Burden of Disease (GBD) study, the global number of cases of non-alcoholic fatty liver disease (NAFLD) was estimated at 48.35 million (95% UI: 44.23 to 52.36 million). In the same year, NAFLD was associated with 138,328 deaths (95% UI: 108,288–173,905) across all age groups, accounting for 0.204% of the total burden from all 369 diseases and injuries analyzed in the GBD study. This reflects a 58.14% increase in NAFLD-related deaths since 1990. Within the central focus of this study—adults aged ≥45 years—the age-standardized prevalence rate (ASPR) reached 30,016.22 per 100,000 population (95% UI: 23,519.09 to 37,290.44) in 2021, underscoring the disproportionately high burden of NAFLD in this age group.

This population exhibited a considerable burden of mortality and disability, with an age-standardized mortality rate (ASMR) of 5.04 per 100,000 (95% UI: 3.44–7.28) and an age-standardized disability-adjusted life year (DALY) rate of 124.7 per 100,000 (95% UI: 83.88–183.07). Furthermore, a pronounced sex-based disparity was observed, with women in this age group consistently exhibiting a higher incidence rate.

### Epidemiological Trends in NAFLD Among Adults Aged ≥ 45 Years

During the period from 1990 to 2021, the ASIR for NAFLD in individuals aged 45 years and older rose by 18.3%, escalating from 557.89 instances for every 100,000 individuals (with a 95% uncertainty interval [UI]: 358.23–804.92) to 660.42 instances (95% UI: 424.09–952.42), corresponding to an estimated annual percentage change (EAPC) value of 0.53. In a similar way, the age-standardized prevalence rate (ASPR) of this demographic demonstrated a marked increase of 24.5%, climbing from 24,115.89 per 100,000 (95% UI: 18,721.86−30,341.58) to 30,016.22 (95% UI: 23,519.09−37,290.44) over the same period, with an EAPC of 0.74. The rising prevalence of NAFLD in older persons is highlighted by these trends, reflecting broader shifts in metabolic risk factors and aging populations globally (Table 1).

Between 1990 and 2021, both the mortality and DALYs associated with NAFLD increased. The age-standardized mortality rate (ASMR) and age-standardized DALYs rate (ASDR) also experienced growth, with ASMR rising from

**Table 1. ASIR and ASPR of NAFLD in 1990 and 2021 for all locations, with EAPC from 1990 and 2021.**

| location | Age-standardized incidence per 100 000 population (95% UI) | | 1990–2021 EAPC of ASIR (95%CI) | Age-standardized Prevalence per 100 000 population (95% UI) | | 1990–2021 EAPC of ASPR (95%CI) |
|---|---|---|---|---|---|---|
| | 1990 | 2021 | | 1990 | 2021 | |
| Global | 557.89(358.23-804.92) | 660.42(424.09-952.42) | 0.53* (0.51-0.56) | 24115.89(18721.86-30341.58) | 30016.22(23519.09-37290.44) | 0.72* (0.65-0.79) |
| **SDI** | | | | | | |
| Low SDI | 614.94(399.61-885.04) | 670.78(438.19-962.48) | 0.28* (0.26-0.31) | 25083.39(19337.63-31711.47) | 27857.03(21531.82-35053.88) | 0.32* (0.28-0.37) |
| Low-middle SDI | 658.34(425.52-946.64) | 733.57(476.7-1058.82) | 0.37 *(0.35-0.39) | 27523.14(21269.35-34683.07) | 31813.65(24855.69-39702.62) | 0.47* (0.43-0.52) |
| Middle SDI | 646.07(418.06-933.44) | 731.04(474.01-1056.8) | 0.36 *(0.34-0.39) | 28453.11(22125.29-35735.24) | 33817.13(26499.76-41925.1) | 0.57 *(0.49-0.65) |
| High-middle SDI | 544.35(352.62-787.52) | 649.97(420.83-944.13) | 0.52* (0.49-0.56) | 24670.77(19189.71-31002.07) | 31166.86(24473.04-38684.42) | 0.75 *(0.65-0.85) |
| High SDI | 390.65(252.61-565.33) | 481.07(312-695.65) | 0.70* (0.67-0.72) | 16648.09(12883.77-21026.91) | 21902.07(17173.36-27286.75) | 0.96 *(0.91-1.01) |
| **Region** | | | | | | |
| High-income North America | 412.65(265.29-596.22) | 498.74(320.3-717.62) | 0.67* (0.64-0.70) | 16162.98(12291.96-20664.1) | 20191.7(15649.99-25463.66) | 0.87* (0.81-0.93) |
| Australasia | 329.87(210.54-480.19) | 385.92(244.91-567.21) | 0.50* (0.47-0.54) | 14506.35(11137.46-18471.09) | 18740.51(14555.32-23490.73) | 0.85* (0.81-0.89) |
| High-income Asia Pacific | 352.55(226.93-513.34) | 401.12(257.83-585.39) | 0.44*(0.37-0.52) | 14899.03(11474.84-18880.22) | 17306.01(13476.47-21847.44) | 0.61* (0.52-0.69) |
| Western Europe | 370.39(238.11-536.69) | 432.06(279.24-624.46) | 0.51* (0.49-0.54) | 16027.14(12456.72-20229.86) | 21426.35(16825.69-26574.8) | 0.99* (0.94-1.04) |
| Southern Latin America | 395.43(250.85-574.71) | 462.71(293.53-673.29) | 0.51 *(0.48-0.55) | 15994.49(12238.79-20417) | 20773.94(15979.18-26395.72) | 0.88* (0.84-0.92) |
| Eastern Europe | 490.24(313.6-711.28) | 515.29(328.69-750.57) | 0.17* (0.16-0.19) | 23391.14(18219.48-29373.79) | 25751.77(20135.14-32042.04) | 0.30* (0.29-0.32) |
| Central Europe | 458.71(292.5-667.75) | 475.98(304-693.56) | 0.12* (0.11-0.13) | 23214.34(18122.95-29045.57) | 25775.13(20217.1-31883.29) | 0.37* (0.35-0.39) |
| Central Asia | 603.51(387.12-875.45) | 643.96(412.54-937.94) | 0.22* (0.20-0.24) | 28844.63(22457.86-36172.25) | 32606.11(25593.43-40548.31) | 0.42* (0.38-0.46) |
| Central Latin America | 656.99(419.77-946.51) | 693.16(443.67-1005.65) | 0.18* (0.17-0.19) | 31139.49(24316.35-38708.19) | 34903.56(27464.89-43173.27) | 0.40 *(0.39-0.41) |
| Andean Latin America | 575.17(365.85-841.55) | 611.03(390.97-888.03) | 0.21* (0.19-0.22) | 27091.73(20999.97-34169.83) | 31268.65(24587.87-38757.71) | 0.51* (0.50-0.53) |
| Caribbean | 622.77(400.06-901.22) | 657.69(421.83-953.5) | 0.19* (0.17-0.20) | 29709.33(23166.18-36958.33) | 33260.86(26173.42-41219.78) | 0.40* (0.39-0.42) |
| Tropical Latin America | 673(430.85-975.9) | 720.44(458.95-1050.57) | 0.24* (0.23-0.26) | 31216.17(24318.59-39330.53) | 34594.38(27070.83-42880.95) | 0.38* (0.37-0.39) |
| East Asia | 613.52(393.97-886.05) | 713.25(458.49-1029.62) | 0.38 *(0.31-0.44) | 26423.6(20359.54-33390.87) | 32367.15(25204.17-40526.4) | 0.64* (0.42-0.86) |
| Southeast Asia | 664.68(428.2-960.46) | 717.49(462.38-1039.37) | 0.26* (0.25-0.28) | 29040.87(22535.23-36601.94) | 32751.37(25594.2-40760.14) | 0.40* (0.39-0.41) |
| Oceania | 649.74(422.48-939.12) | 684(444.62-979.39) | 0.19* (0.16-0.21) | 29298.18(22674.06-37001.06) | 32862.49(25581.1-41072.85) | 0.39 *(0.36-0.42) |
| North Africa and Middle East | 862.43(549.21-1259.37) | 978(621.16-1421.17) | 0.46*(0.44-0.48) | 42928.76(33837.64-52848.3) | 54017.08(43474.71-64718.42) | 0.80* (0.76-0.85) |
| South Asia | 638.81(410.37-916.08) | 720.59(462.95-1034.68) | 0.39* (0.36-0.43) | 25188.81(19306.15-32028.34) | 28912.4(22274.07-36524.24) | 0.44* (0.36-0.52) |
| Southern Sub-Saharan Africa | 643.88(414.74-932.24) | 671.85(434.11-974.11) | 0.14 *(0.13-0.16) | 28114.12(21712.66-35311.65) | 32055.8(25045.22-39858.7) | 0.41* (0.39-0.44) |
| Western Sub-Saharan Africa | 615.79(395.3-887.7) | 668.62(430.11-970.06) | 0.27* (0.25-0.29) | 26256.96(20314.83-33178.03) | 29443.43(22815.54-37051.38) | 0.35* (0.33-0.37) |
| Eastern Sub-Saharan Africa | 570.92(367.46-821.54) | 613.4(395.7-885.93) | 0.24* (0.23-0.25) | 23581.24(18208.79-29822.45) | 26290.21(20343.83-33146.81) | 0.34* (0.32-0.36) |
| Central Sub-Saharan Africa | 528.16(339.63-763.11) | 556.34(358.55-799.31) | 0.16* (0.14-0.19) | 21795.1(16628.82-27750.95) | 23663.89(18182.73-30060.89) | 0.28* (0.25-0.32) |

ASIR Age-standardized incidence rate, ASPR Age-standardized prevalence rate, EAPC Estimated annual percentage change, CI Confidence interval, SDI Socio-demographic index, UI Uncer-tainty interval. * Statistically significant (P < 0.05).

4.81 per 100,000 population (95% UI: 3.12–7.29) in 1990 to 5.04 per 100,000 population (95% UI: 3.44–7.28) in 2021 (EAPC = 0.17). In 2021, the ASDR increased to 124.7 per 100,000 people (95% UI: 83.88–183.07) from 120.05 per 100,000 people in 1990 (95% UI: 77. 24–183. 21) (EAPC=0.12) (Table 2).

At the national level in 2021, among people older than 45, With an ASIR of 1063.2 per 100,000 individuals, Afghanistan had the highest rate of NAFLD.(95% UI: 693.4–1532.7), for which Iran (Islamic Republic of) (1043 per 100,000 population; 95% UI: 664.7–1521.3) and Libya (1034.2 per 100,000 population; 95% UI: 652.7–1514.8) followed. With an ASIR of 337.7 per 100,000 individuals (95% UI: 218.1–487.2), Denmark had the lowest rate of NAFLD, followed by Japan (372.3 per 100,000 population; 95% UI: 239.6–542.8) and Switzerland (372.4 per 100,000 population; 95% UI: 238.4–541.2). (Fig 1A). Kuwait, Egypt, Iran (Islamic Republic of), Qatar and Saudi Arabia presented the highest ASPR of NAFLD (61,255.9 out of every 100,000 people, with a 95% uncertainty inter- val of 50,233.5 to 71,986.3; 59,912.1 out of every 100,000 people, with a 95% uncertainty interval of 49,063.1 to 70,454; 59,428 out of every 100,000 people, 95% UI: 48217.4–70660.7; 59300.4 per 100,000 population, 95% UI: 48761.8–69829.5 and 57345.1 per 100,000 population, 95% UI: 46337.1–68462.6). Japan, Denmark and Finland demonstrated the lowest ASPR of NAFLD (16136.2 per 100,000 population, 95% UI: 12529.7–20391.7; 16347 per 100,000 population, 95% UI: 12681.7–20531.5; 16757 per 100,000 population, 95% UI: 12847.3–21450.1) (Fig 1B). In 2021, The highest ASMR of NAFLD was recorded in Egypt and Mongolia, with rates of 30.4 and 27.3 per 100,000 population, respectively, and 95% UI ranges of 17.8–49 and 16.8–43. Papua New Guinea and Sri Lanka presented the lowest ASMR of NAFLD (1.4 per 100,000 population, 95% UI: 0.6–2.8, 1.7 per 100,000 population, 95% UI: 0.9–3.1)(Fig 1C). Egypt and Mexico recorded the highest ASDR for NAFLD (643.2 per 100,000 population, 95% UI: 378.9–1030.6, 605.3 per 100,000 population, 95% UI: 376–912.9). Singapore and Papua New Guinea presented the lowest ASDR for NAFLD (34.8 per 100,000 population, 95% UI: 21.9–53.8, 35.4 per 100,000 population, 95% UI:16.6–73.5).(Fig 1D).

Fig 2 displays the actual global and regional ASIR, ASPR, ASMR, and ASDR in comparison to the anticipated levels for each area based on the SDI, presented as yearly data from 1990 to 2021.

Between 1990 and 2021, age-standardized incidence rates (ASIR) and prevalence rates (ASPR) of NAFLD across 21 GBD regions demonstrated a consistent upward trajectory. Notably, both metrics exhibited a bell-shaped correlation with the SDI, peaking in regions with intermediate SDI levels (Figs 2A and 2B). Over the study period, areas classified as medium SDI consistently reported the highest ASIR and ASPR values, while high- and low-SDI regions displayed compar- atively lower rates. This phenomenon was most prominent in North Africa and the Middle East, where NAFLD incidence and prevalence remained persistently elevated, surpassing all other regions globally.

The ASMR and ASDR of NAFLD worldwide and in 21 regions showed a significant nonlinear association with SDI (Figs 2C and 2D). In the low SDI region, ASMR and ASDR rose slowly. Notably, within intermediate SDI territories, Andean Latin America recorded the most severe burden, with age-standardized DALY rates (ASDR) reaching 372.44 per 100,000 (95% UI: 212.12–604.47) in 1990, escalating to 444.89 per 100,000 (95% UI: 255.72–694.22) by 2021. Central Latin America followed closely, demonstrating ASDR values of 345.15 per 100,000 (95% UI: 200.68–552.59) in 1990 and 416.33 per 100,000 (95% UI: 257.22–622.60) in 2021. These figures consistently surpassed projected thresholds derived from SDI-based epidemiological models throughout the three-decade period. Among women, NAFLD in the Andean Latin America region has consistently exhibited a relatively high ASMR with a yearly increasing trend (Fig 3B). Meanwhile, women in Andean Latin America have the highest rates of ASMR and ASDR in NAFLD (Figs 4A and 4B). Between 1990 and 1998, the ASMR of NAFLD of men in the Southern Sub-Saharan African region seen a significant increase, followed by a trend toward stabilization followed by a tendency toward stabilization (Fig 3C). In high-SDI regions, ASDR and mortality rates ASMR generally remained low, yet distinct patterns emerged across nations. A pronounced decline in ASDR was observed in affluent Asia-Pacific nations and Western Europe, contrasting with rising trends in other areas (Fig 3A). Notably, The most rapid rise occurred in Eastern Europe, where

**Table 2. ASMR and ASDR of NAFLD in 1990 and 2021 for all locations, with EAPC from 1990 and 2021.**

| location | Age-standardized Mortality per 100 000 population (95% UI) | | 1990–2021 EAPC of Mortality (95%CI) | Age-standardized DALYs per 100 000 population (95% UI) | | 1990–2021 EAPC of DALYs (95%CI) |
|---|---|---|---|---|---|---|
| | 1990 | 2021 | | 1990 | 2021 | |
| Global | 4.81(3.12-7.29) | 5.04(3.44-7.28) | 0.17* (0.11-0.22) | 120.05(77.24-183.21) | 124.7(83.88-183.07) | 0.12* (0.06-0.19) |
| **SDI** | | | | | | |
| Low SDI | 5.98(3.67-9.56) | 5.44(3.62-8.13) | −0.37* (−0.42–0.33) | 144.02(89.47-228.33) | 127.81(84.34-193.18) | −0.48* (−0.53–0.44) |
| Low-middle SDI | 5.26(3.12-8.64) | 5.89(3.79-8.85) | 0.41* (0.38-0.43) | 124.23(74.28-202.8) | 142.62(91.33-216.2) | 0.50* (0.46-0.53) |
| Middle SDI | 5.01(3.27-7.58) | 5.5(3.78-7.85) | 0.38* (0.34-0.41) | 123.15(79.94-186.54) | 133.5(90.58-193.06) | 0.28* (0.24-0.31) |
| High-middle SDI | 4.66(3.03-6.93) | 4.25(2.86-6.16) | −0.30* (−0.45–0.16) | 115.52(74.35-174.01) | 107.24(70.89-158.62) | −0.26* (−0.46–0.07) |
| High SDI | 4.53(2.92-6.85) | 4.71(3.19-6.77) | 0.16* (0.07-0.26) | 116.99(74.28-179.68) | 119.44(79.55-174.4) | 0.11 (−0.01-0.22) |
| **Region** | | | | | | |
| High-income North America | 3.36(2.14-5.14) | 4.99(3.37-7.26) | 1.55* (1.40-1.70) | 88.6(55.12-138.99) | 129.37(85.58-192.41) | 1.56* (1.40-1.71) |
| Australasia | 2.49(1.57-3.82) | 4.05(2.81-5.67) | 1.80* (1.65-1.96) | 66.33(41.17-103.22) | 103.27(70.84-144.98) | 1.75* (1.58-1.92) |
| High-income Asia Pacific | 5.08(3.57-7.14) | 2.75(1.86-3.88) | −2.20* (−2.37–2.03) | 122.66(86.24-173.53) | 59.8(40.52-85.28) | −2.58* (−2.75–2.41) |
| Western Europe | 6.88(4.19-10.57) | 5.59(3.67-8.02) | −0.69* (−0.85–0.53) | 176.99(105.59-276.85) | 141.24(91.32-205.82) | −0.74* (−0.93–0.55) |
| Southern Latin America | 5.68(3.15-9.5) | 5.53(3.32-8.74) | 0.32* (0.20-0.45) | 149.95(82.23-254.44) | 140.12(82.49-225.6) | 0.21* (0.08-0.35) |
| Eastern Europe | 2.94(1.82-4.68) | 7.56(4.52-12.11) | 3.11* (2.55-3.67) | 79.2(48.18-128.14) | 229.15(133.35-374.36) | 3.36* (2.69-4.04) |
| Central Europe | 4.34(2.66-6.84) | 5.44(3.33-8.57) | 0.45 *(0.30-0.59) | 114.2(68.2-183.86) | 147.54(88.2-237.1) | 0.49* (0.31-0.66) |
| Central Asia | 6.85(4.26-10.68) | 10.39(6.27-16.42) | 1.54* (1.32-1.77) | 171.85(105.92-269.71) | 258.51(153.74-416.4) | 1.42* (1.20-1.64) |
| Central Latin America | 13.2(7.82-20.7) | 15.84(9.94-23.26) | 0.58* (0.50-0.66) | 345.15(200.68-552.59) | 416.33(257.22-622.6) | 0.55* (0.46-0.65) |
| Andean Latin America | 14.9(8.65-23.9) | 18.86(11.02-29.02) | 0.77* (0.68-0.85) | 372.44(212.12-604.47) | 444.89(255.72-694.22) | 0.51* (0.41-0.60) |
| Caribbean | 8.79(5.14-14.08) | 9.42(5.6-14.76) | 0.17 (−0.07-0.42) | 218.69(124.98-356.76) | 242.59(141.49-388.59) | 0.30 *(0.05-0.55) |
| Tropical Latin America | 3.53(2.13-5.62) | 4.39(2.77-6.59) | 0.99* (0.87-1.10) | 94.19(55.67-152.94) | 116.45(72.05-178.66) | 0.92* (0.80-1.03) |
| East Asia | 3.26(2.25-4.67) | 2.59(1.82-3.58) | −0.53* (−0.65–0.41) | 81.69(56.32-117.15) | 60.6(42.39-84.14) | −0.82* (−0.94–0.70) |
| Southeast Asia | 5.07(3.08-8.27) | 5.78(3.76-8.58) | 0.44* (0.37-0.51) | 123.5(75.99-198.66) | 134.63(87.28-201.95) | 0.27* (0.21-0.33) |
| Oceania | 2.77(1.51-5.12) | 2.51(1.49-4.08) | −0.48 *(−0.60–0.37) | 71.77(38.92-132.23) | 64.14(37.53-105.7) | −0.53* (−0.65–0.40) |
| North Africa and Middle East | 8.25(4.62-14.06) | 8.6(5.38-13.38) | 0.18* (0.11-0.24) | 169.58(96.83-284.57) | 188.35(117.79-293.44) | 0.42* (0.39-0.45) |
| South Asia | 3.41(2.03-5.72) | 4(2.6-6.09) | 0.47* (0.42-0.53) | 87.56(52.31-145.57) | 97.51(62.91-149.24) | 0.28* (0.24-0.33) |
| Southern Sub-Saharan Africa | 5.42(3.21-8.94) | 8.57(5.98-12.22) | 1.27* (0.83-1.72) | 132.85(79.03-218.82) | 209.37(144.28-304.86) | 1.30* (0.85-1.75) |
| Western Sub-Saharan Africa | 8.74(5.07-14.4) | 8.64(5.75-12.78) | −0.09* (−0.13–0.05) | 202.17(118.27-332.45) | 195.48(128.5-293.22) | −0.16* (−0.20–0.12) |
| Eastern Sub-Saharan Africa | 7.26(4.58-11.28) | 7.56(4.88-11.61) | −0.01 (−0.06-0.05) | 172.84(109.48-268.35) | 175.18(112.88-270.73) | −0.11* (−0.18–0.05) |
| Central Sub-Saharan Africa | 5.53(2.9-10.06) | 5.01(2.64-9.16) | −0.45* (−0.53–0.37) | 142.02(75.72-255.52) | 126.64(67.56-230.86) | −0.50* (−0.58–0.41) |

EAPC Estimated annual percentage change, DALYs Disability-adjusted life years, CI Confidence interval, SDI Socio-demographic index, UI Uncer-tainty interval. * Statistical-ly significant (P < 0.05).

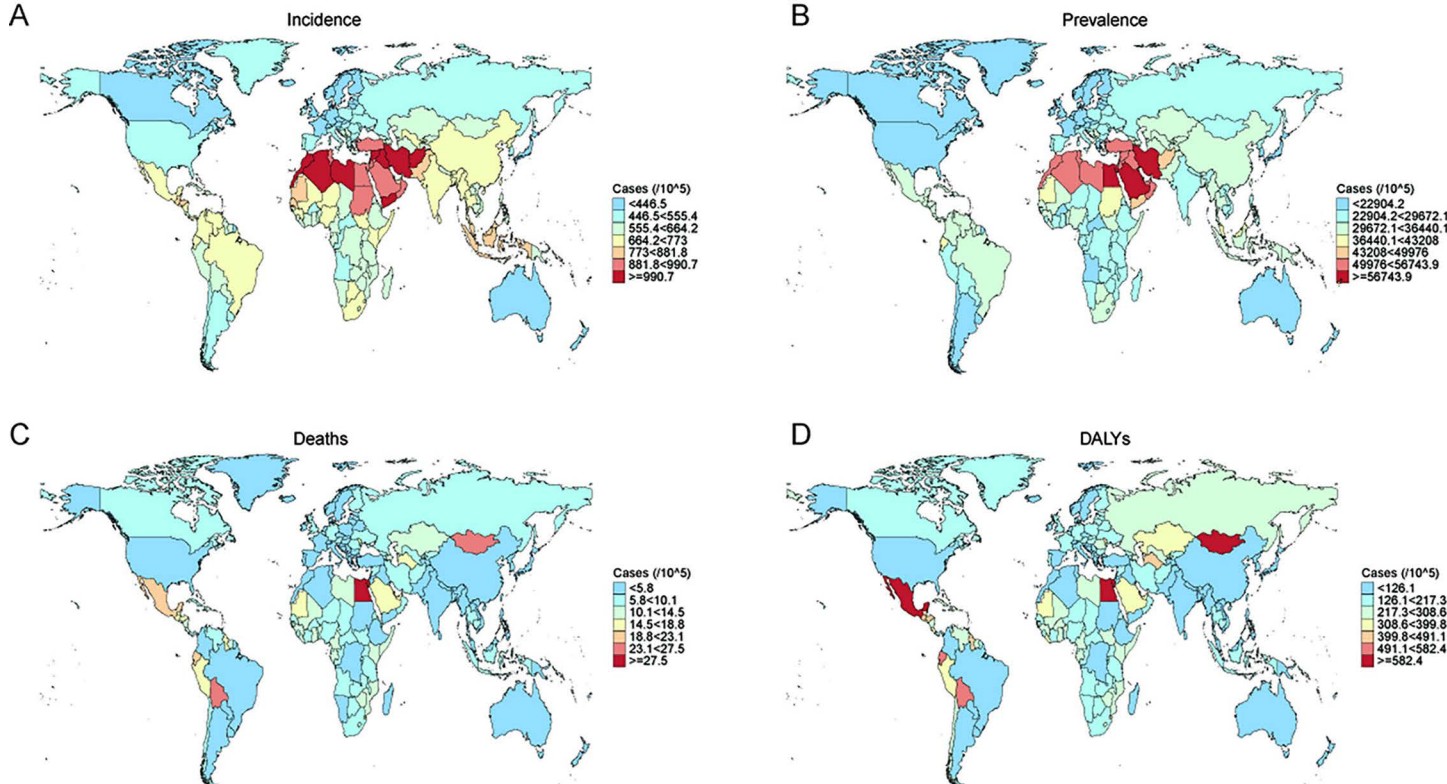

**Fig 1. Global Burden of NAFLD in Adults Aged ≥45 Years (2021) A: Age-standardized incidence rates (per 100,000 population).** Highest rates in Afghanistan, Iran, and Libya; lowest in Denmark, Japan, and Switzerland. B: Age-standardized prevalence rates (per 100,000 population). Highest in Kuwait, Egypt, Iran, Qatar, and Saudi Arabia; lowest in Japan, Denmark, and Finland. C: Age-standardized mortality rates (per 100,000 population). Highest in Egypt and Mongolia; lowest in Papua New Guinea and Sri Lanka. D: Age-standardized DALY rates (per 100,000 population). Highest in Egypt and Mexico; lowest in Singapore and Papua New Guinea. Abbreviations: NAFLD, Nonalcoholic Fatty Liver Disease; DALY, Disability-Adjusted Life Year. Uncertainty intervals (95% UI) denote data variability. Basemap source: Natural Earth (public domain, https://www.naturalearthdata.com/). Maps were generated using the rnaturalearth package in R.

the EAPC in ASDR was 3.36 (95% CI: 2.69–4.04) between 1990 and 2021, surpassing trends in all other regions. This divergence highlights the heterogeneous impact of socioeconomic development on NAFLD-related mortality, even within comparable development tiers.

## Cross-national NAFLD health inequality

In 1990 and 2021, the SII (for every 100,000 individuals) for DALYs were −0.39 and −0.33 showing an inverse relationship between age-adjusted DALY rates and the SDI index (see Fig 5). The trends we see show a steady decrease in the differences of age-adjusted NAFLD rates between wealthy and poorer countries from 1990 to 2021. During this period, concentration index for DALYs and mortality exhibited progressive reductions, with values for DALYs rising from −0.13 (1990) to −0.10 (2021), and mortality indices shifting from −0.15 (1990) to −0.12 (2021) (Figs 6C and 6D). Conversely, prevalence and incidence displayed opposing patterns, as their concentration index transitioned from neutral or positive values to negative territory (prevalence: 0 [1990] to −0.06 [2021]; incidence: 0.02 [1990] to −0.04 [2021]). These divergent trajectories suggest that while mortality-related inequalities diminished, disparities in disease onset and persistence became more pronounced over time (Figs 6A and 6B).

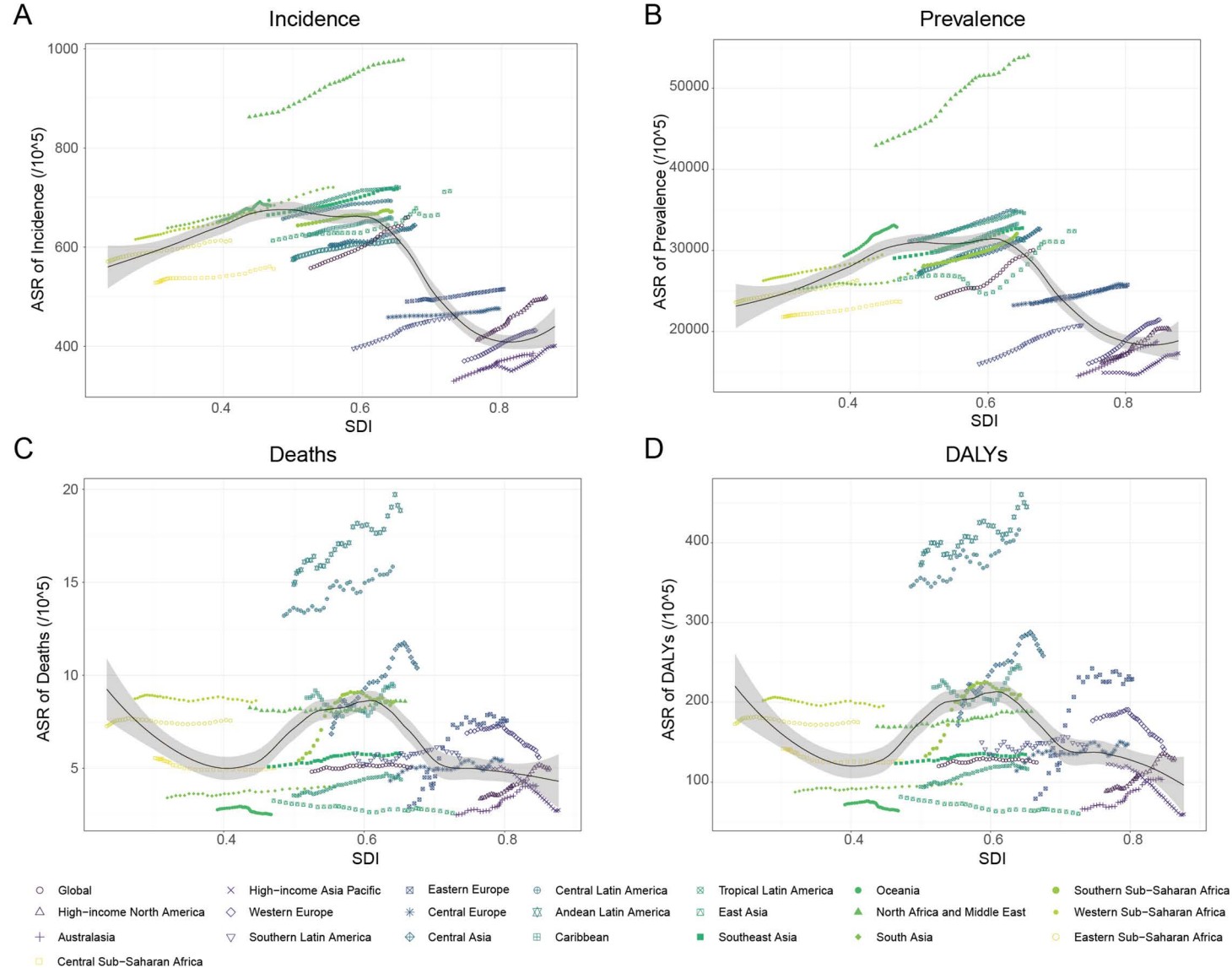

**Fig 2. Trends in Age-Standardized NAFLD Rates by SDI Quintiles and GBD Regions (1990–2021) A: Incidence rates showing a bell-shaped SDI correlation, peaking in medium-SDI regions (e.g., North Africa/Middle East).** B: Prevalence rates mirroring incidence trends, with highest burdens in medium-SDI regions. C: Mortality rates exhibiting nonlinear SDI associations, highest in Andean and Central Latin America. D: DALY rates highlighting severe burdens in Andean Latin America and Central Latin America. Abbreviations: SDI, Socio-demographic Index; GBD, Global Burden of Disease; ASIR, Age-Standardized Incidence Rate; ASPR, Age-Standardized Prevalence Rate; ASMR, Age-Standardized Mortality Rate; ASDR, Age-Standardized DALY Rate.

## Bayesian age-period-cohort (BAPC) model prediction

The natural history from early fat accumulation in the liver to severe scarring, liver cirrhosis, and hepatocellular carcinoma (HCC) often spans decades [2]. Individuals currently aged ≥ 45 years represent a critical cohort entering the peak risk period for disease progression and complications over the next 10–15 years [6]. Projections to 2035 allow us to assess the anticipated burden within this high-risk timeframe, capturing the trajectory of prevalent cases diagnosed today as they age and the emergence of incident cases within this vulnerable demographic. This timeframe aligns with

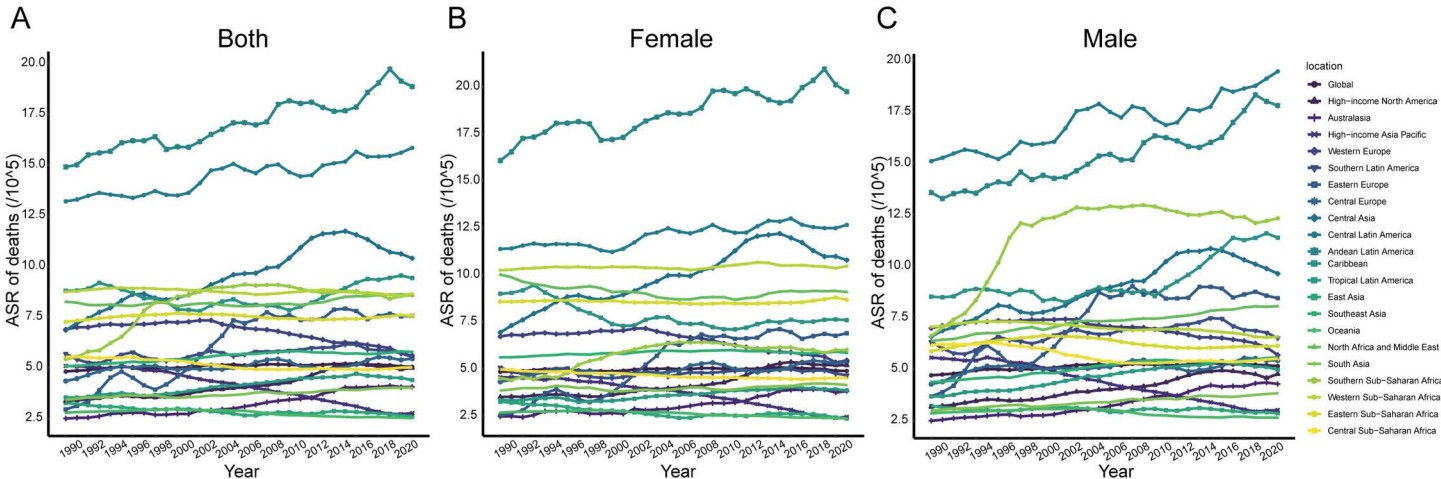

**Fig 3. Sex-Stratified Trends in Age-Standardized DALY Rates (1990–2021) A: Both sexes.** Eastern Europe showed the steepest increase (EAPC = 3.36). B: Females. Andean Latin America had the highest rates with rising trajectories. C: Males. Southern Sub-Saharan Africa experienced sharp increases before stabilizing post-1998. Abbreviations: EAPC, Estimated Annual Percentage Change. Shaded areas represent 95% uncertainty intervals.

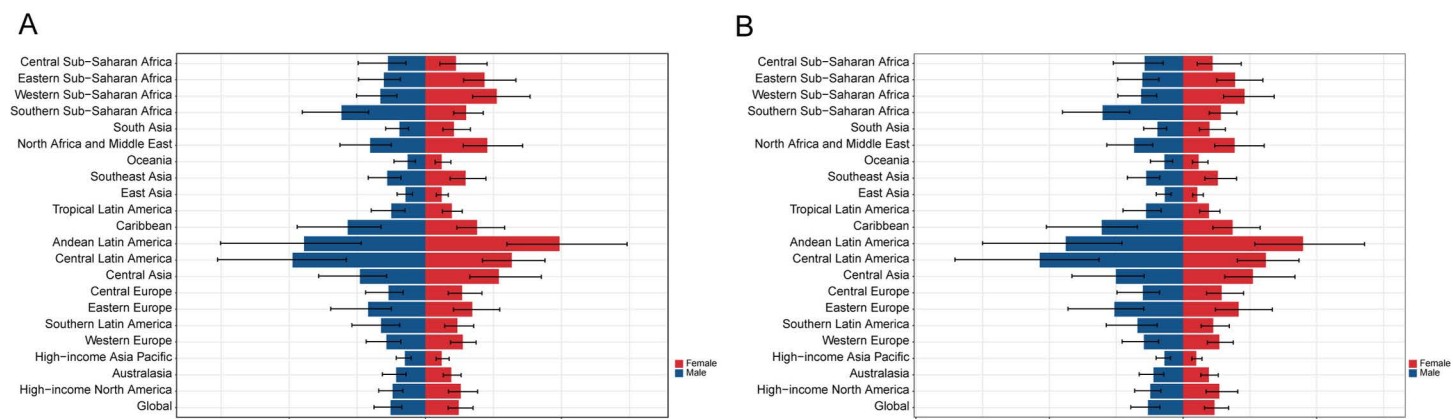

**Fig 4. Sex-Specific Mortality and DALY Rates by Region (2021) A: Age-standardized mortality rates (ASMR).** Andean Latin American women bore the highest burden. B: Age-standardized DALY rates (ASDR). Andean Latin America and North Africa/Middle East had the highest rates, with women disproportionately affected. Regional order: Ranked from highest to lowest burden. GBD regions are grouped by SDI levels.

strategic public health planning cycles and is crucial for evaluating the long-term impact of current interventions and demographic shifts.

Using Bayesian ageperiod cohort models, we forecast ASIR, ASMR, and ASDR in people over 45 from 2021 to 2035, with different trends seen between genders, in order to understand the trends of NAFLD in this age group after 2021.

From 1990 to 2021, the ASIR among women has consistently been higher than that among men. The rate of ASIR for women is always greater than that for men during the forecasted time, showing that women face a heavier load of NAFLD. By 2035, the ASIR of NAFLD is expected to increase to 826.11 per 100,000 women and 665.72 per 100,000 men. For women, the projected incidence rates demonstrate a steady upward trend over time, with a more substantial increase anticipated as the timeline advances toward 2035.(Figs 7A and 7B).

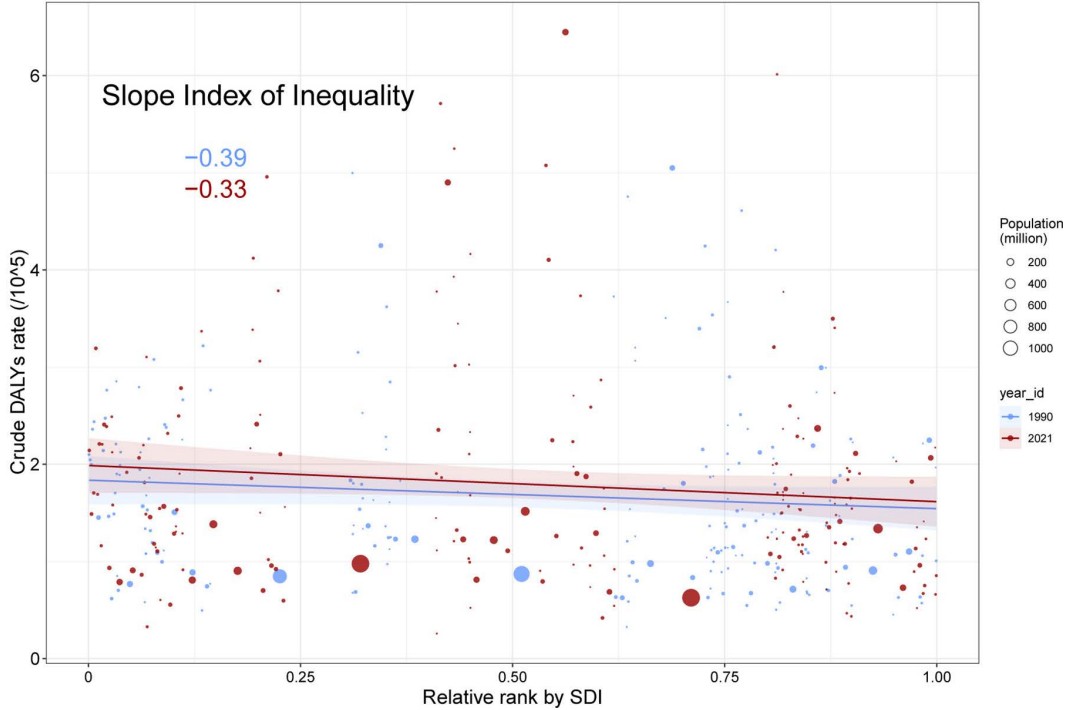

**Fig 5. Health Inequality in DALY Rates by SDI (1990 vs. 2021).** Slope Index of Inequality (SII) for age-standardized DALY rates. Negative SII values (−0.39 in 1990; −0.33 in 2021) indicate higher burdens in low-SDI regions. The narrowing gap reflects reduced inequality between high- and low-income nations. Abbreviations: SII, Absolute difference in DALY rates between extreme SDI percentiles.

Epidemiological surveillance data from 1990 through 2021 demonstrate sustained annual growth in ASMR, with longitudinal analyses revealing persistent gender-specific differentials-male mortality consistently surpassing female rates throughout this observation window. Modeling projections suggest a paradigm shift during the 2021–2035 period, with both genders exhibiting transition to declining trajectories. Quantitative estimates predict that female ASMR will decline to 4.34 cases per 100,000 population by 2035 (representing an 11.97% reduction relative to 2021 baseline values), while male rates decrease to 4.80 per 100,000 (9.60% decline over the same timeframe). Notably, comparative analysis reveals a more substantial mortality rate reduction in female populations, with their projected percentage decrease exceeding male counterparts by 2.37 percentage points (Figs 7C and 7D).

From 1990 to 2021, ASDR tended to be stable, but the ASDR for men was consistently greater than that for women. Predictions indicate that from 2021 to 2035, the ASDR of both men and women will show a downward trend. By 2035, the ASDR of female NAFLD will drop to 105.32 per 100,000 population, a decrease of 10.88% compared with 118.18 in 2021. The ASDR of male NAFLD dropped to 120.54 per 100,000 population, a decrease of 9.98% compared with 133.91 in 2021. The decline of ASDR in women is more obvious (Figs 7E and 7F).

### Decomposition analysis of incidence

According to the decomposition analysis, globally, the impacts of population growth, disease pattern transformation and population aging on the occurrence of NAFLD are 58.68%, 33.76% and 7.56% respectively. Among adults aged 45 years and above, the expansion of the population size has emerged as the predominant driver behind the rising prevalence of NAFLD, accounting for 81.97% of the observed increase in cases. Concurrently, changes in disease incidence patterns

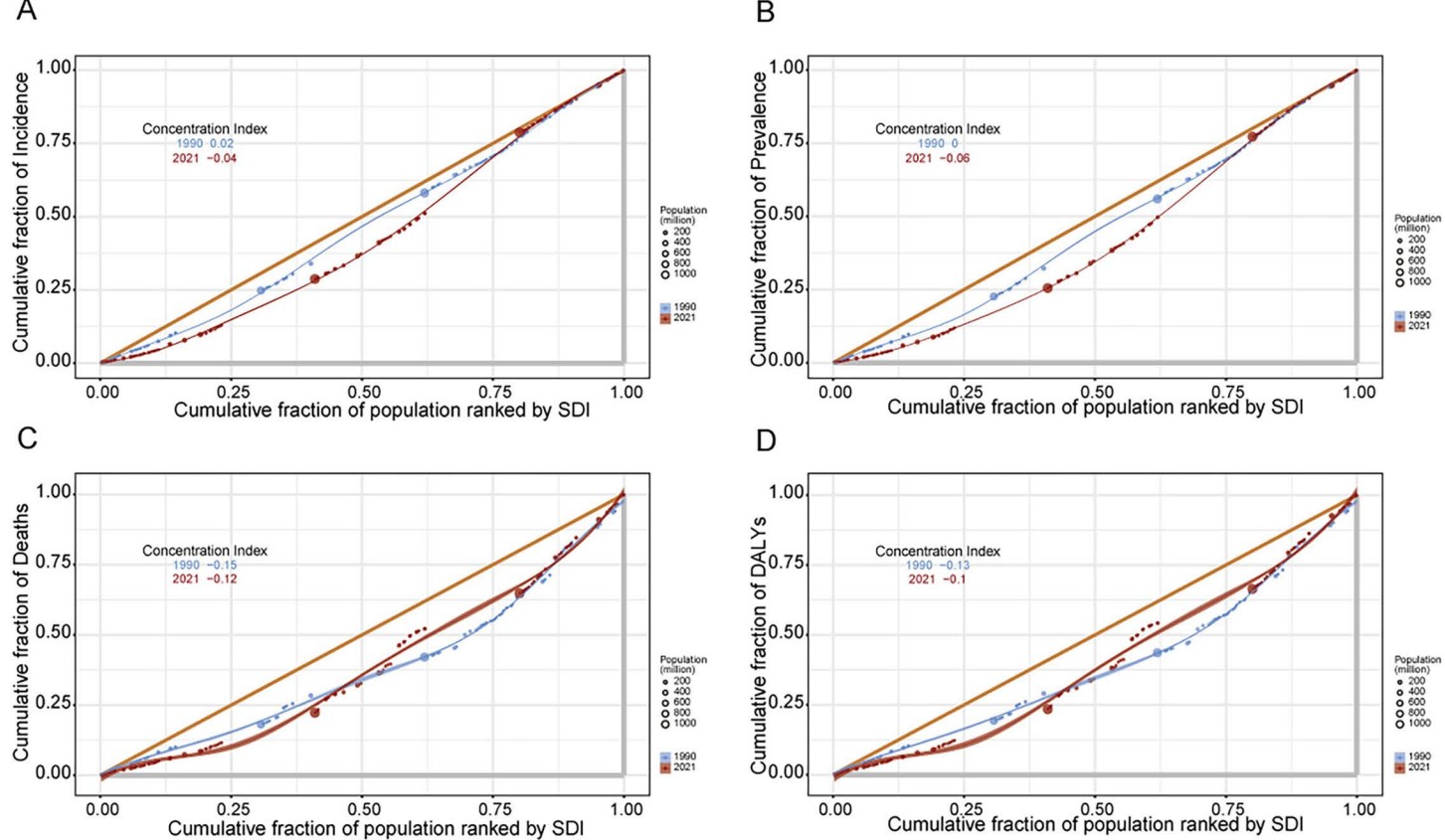

**Fig 6. Concentration Indices (CI) for NAFLD Burden Metrics (1990 vs. 2021).** A: Incidence CI shifted from 0.02 (1990) to −0.04 (2021), indicating growing disparities in disease onset. B: Prevalence CI changed from 0 (1990) to −0.06 (2021), reflecting widening inequalities in disease persistence. C: Mortality CI improved from −0.15 (1990) to −0.12 (2021), signaling reduced mortality inequality. D: DALY CI rose from −0.13 (1990) to −0.10 (2021), showing modest reductions in disability inequality. Abbreviations: Negative CI, Concentration of burden in low-SDI populations.

contributed to 18.97% of this growth. Interestingly, the global aging phenomenon demonstrated a moderating effect, with a negative contribution of −0.63% to the overall rise in NAFLD incidence. (Figs 8A and 8B)

## Discussion

Our analysis reveals that NAFLD burden among adults aged ≥ 45 years has increased substantially (63% growth) compared to general population trends (38%), reflecting the synergistic impact of demographic aging and the expanding metabolic syndrome pandemic [19]. Age-related physiological alterations—including diminished hepatic β-oxidation, mitochondrial dysfunction, and chronic inflammation-accelerate fibrosis progression by 4–5% per decade after age 40 [5], culminating in a 15-fold elevated HCC risk by age 65 [6]. This is compounded by metabolic multimorbidity in aging populations, where >70% of NAFLD patients ≥ 45 years concurrently exhibit T2DM, hypertension, or CVD [7], creating a bidirectional disease cascade that amplifies liver damage.

Whereas high-income nations have achieved modest mortality reductions through enhanced detection and therapeutic advances, low to middle-income countries face a dual challenge of rising incidence rates compounded by limited healthcare access [20]. Notably, the Socio-demographic Index (SDI) reveals a critical nonlinear relationship with NAFLD burden: regions with medium SDI (e.g., North Africa, Middle East, Latin America) exhibit the highest incidence and DALY rates

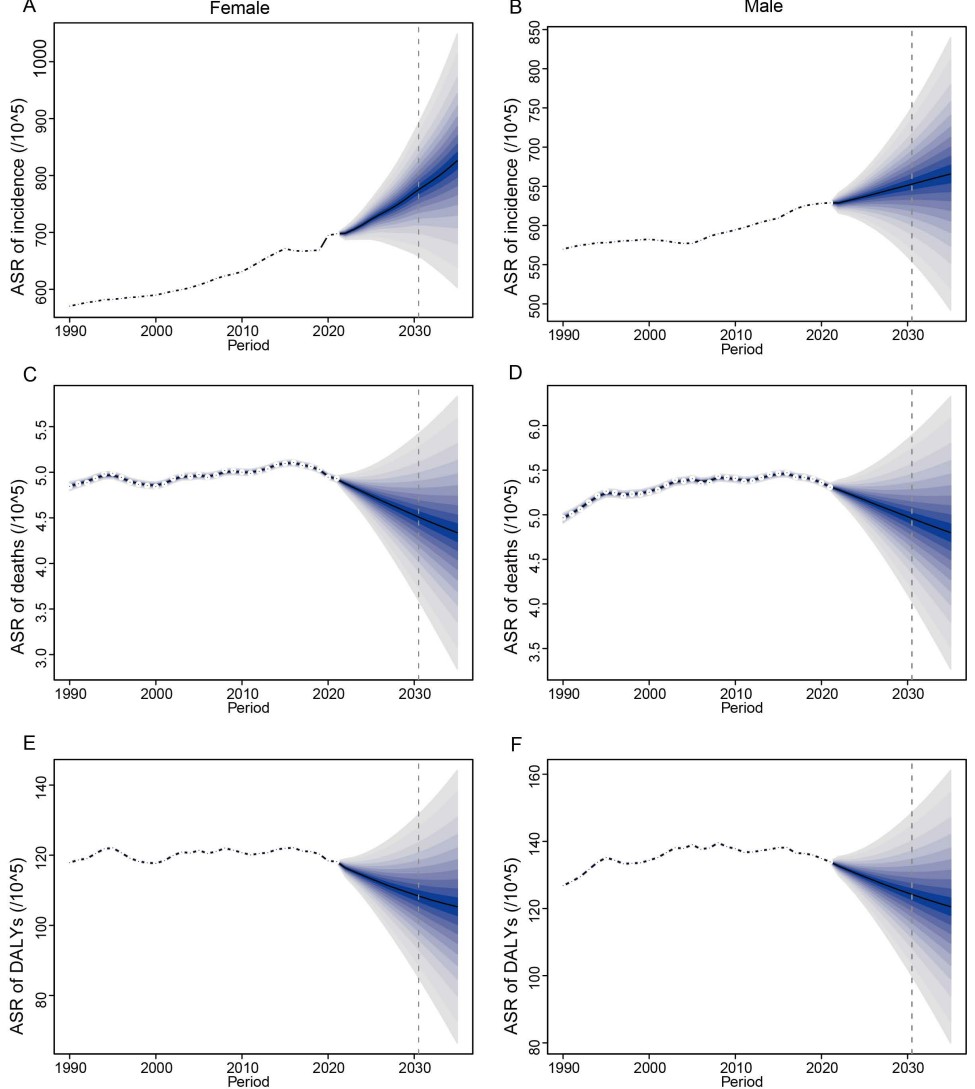

**Fig 7. BAPC Projections of NAFLD Burden to 2035.** A–B: Incidence rates projected to rise, with women consistently higher (826.11 vs. 665.72 per 100,000 by 2035). C–D: Mortality rates declining more sharply in women (11.97% reduction) than men (9.60%). E–F: DALY rates decreasing, with steeper declines in women (10.88% vs. 9.98% in men). Abbreviations: BAPC, Bayesian age-period-cohort modeling. Dashed lines represent Projections; shaded areas represent 95% uncertainty intervals.

(Figs 2A-D), surpassing both high- and low-SDI areas. This "SDI paradox" arises from rapid urbanization in transitioning economies, driving nutrition transitions towards Western diets and sedentary lifestyles [21], while healthcare systems remain under-resourced for early NAFLD detection and metabolic management. In contrast, high-SDI regions benefit from established prevention programs, yet face emerging challenges in aging subgroups and socioeconomic disparities within countries (e.g., Eastern Europe's 3.36% annual ASDR rise).

Our projections indicate potential stabilization of mortality rates by 2035, although absolute case numbers will continue escalating due to persistent demographic shifts. By 2050, the global population aged ≥60 years will double [9], with low/middle-income countries experiencing the most rapid aging. This demographic wave will intersect with SDI-driven risk factor disparities: in high-SDI nations, aging populations may offset mortality gains from advanced care, whereas

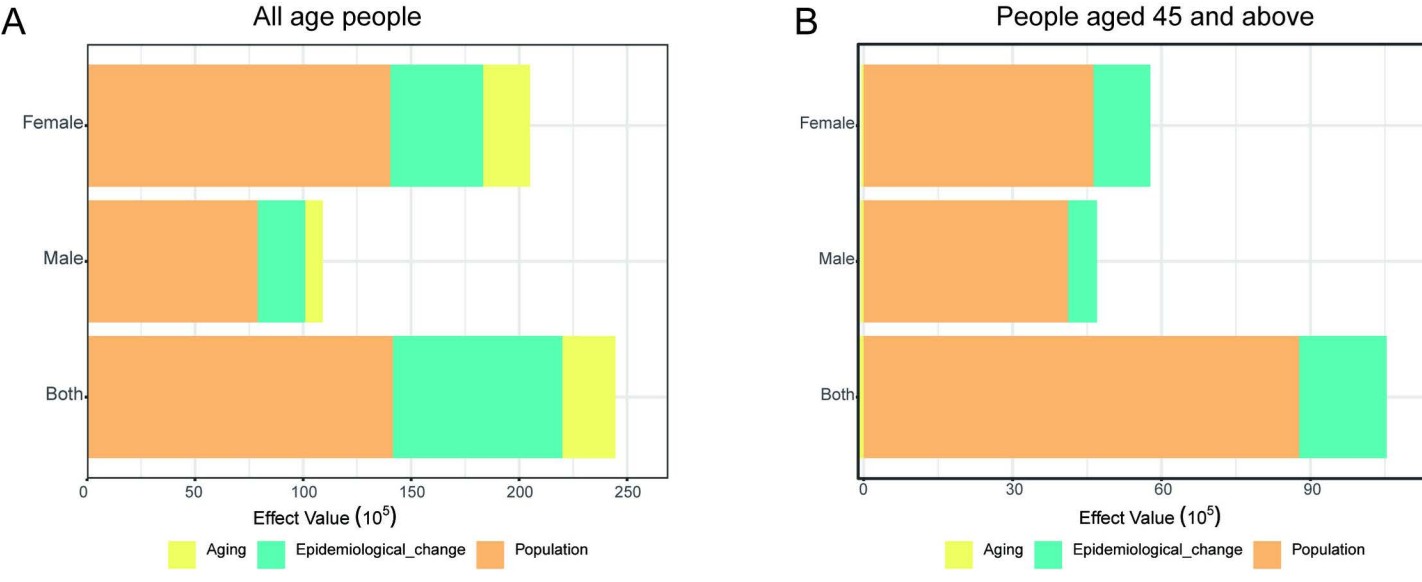

**Fig 8. Decomposition Analysis of NAFLD Incidence Drivers (1990–2021).** A: Global contributions to incidence change.population growth (58.68%) and Epidemiological shifts (33.76%) were primary drivers; aging had minimal impact (7.56%). B: Contributions to the rise in NAFLD prevalence among adults aged ≥45 years: Population growth, Epidemiological shifts and aging accounted for 81.97%, 18.97% and −0.63% respectively.

medium-SDI regions confront exponential growth in NAFLD-related complications due to delayed diagnosis and limited fibrosis monitoring.

The disproportionate disease burden observed in Middle Eastern and Latin American populations aligns with regional epidemics of obesity (≥ 30%) and type 2 diabetes (≥ 15%) [22]. In Andean Latin America, the confluence of genetic predisposition (PNPLA3 polymorphism frequency: 49%), accelerated population aging, postmenopausal metabolic changes in women, and nutritional transitions explains the exceptionally high DALY rates [23].

Decomposition of contributions to NAFLD incidence reveals distinct patterns in the ≥45-year-old population. Population growth emerged as the dominant driver, accounting for 81.97% of the increase—a contribution greater than that observed in the all-age population. Concurrently, the contribution of epidemiological changes (i.e., age-specific incidence rates) declined from 33.76% to 18.97%, suggesting a more rapid rise in incidence among younger segments of the population. Interestingly, population aging exhibited a significant negative contribution (−0.63%). This could be attributed to the 'healthy survivor effect,' where longer-lived individuals are metabolically healthier, and the progression of existing NAFLD cases to advanced liver disease, thus removing them from the incident pool.

Our findings carry several important implications for clinical practice and public health policy. Firstly, among adults aged ≥45 years, the incidence and prevalence of NAFLD have increased markedly. This rise, coupled with the pronounced sex-specific disparity, characterized by higher incidence in women yet persistently elevated mortality in men, called for differentiated prevention and management strategies. The disproportionately high burden in medium-SDI regions further underscores the need for regionally tailored public health interventions. Secondly, these results highlight the importance of early and systematic liver disease screening in middle-aged and older adults, especially those with metabolic comorbidities. The accelerated progression of fibrosis after age 45 supports the incorporation of non-invasive fibrosis assessment (e.g., FIB-4, ELF test) into routine practice for high-risk patients. Furthermore, sex-specific management protocols should be developed. For women ≥45 years, screening programs should intensify around the menopausal transition, with consideration of hormone replacement therapy's potential hepatoprotective effects in appropriate candidates [24]. For men, aggressive cardiovascular risk factor modification takes priority, given their elevated mortality risk. This includes intensive

lipid management, blood pressure control, and diabetes optimization, as cardiovascular disease remains the leading cause of death in male NAFLD patients [6]. Finally, healthcare systems in high-burden regions require structural adaptations. Integration of NAFLD care into primary healthcare delivery, training of community health workers in basic hepatic steatosis assessment, and development of telehealth platforms for specialist consultation can improve access to care in resource-limited settings [25]. Public health interventions should prioritize policy-level changes, including taxation of sugar-sweetened beverages, urban planning promoting physical activity, and food security programs ensuring access to nutrient-dense foods [26].

This study represents the first comprehensive, age-stratified analysis of global NAFLD burden utilizing the complete GBD 2021 dataset with advanced Bayesian modeling techniques. Our methodological approach offers several advantages: First, harmonized diagnostic criteria across diverse healthcare systems through the FLI surrogate approach, validated against local imaging cohorts. Second, rigorous uncertainty quantification through 1000 Bayesian posterior draws. Third, comprehensive decomposition analysis elucidating the relative contributions of demographic versus epidemiological drivers. Fourth, sex-stratified projections extending to 2035, enabling healthcare planning for anticipated demographic transitions.

Several limitations should be considered in this study. First, diagnostic heterogeneity across countries may introduce systematic bias. The FLI surrogate exhibits suboptimal sensitivity in lean NAFLD populations, particularly in Asian countries where genetic variants (PNPLA3, TM6SF2) increase steatosis risk independent of BMI [27]. This may lead to underestimation of disease burden in populations with high genetic susceptibility but lower average BMI. Second, the ecological nature of our analysis precludes individual-level causal inferences. While we observe associations between SDI levels and disease burden, these relationships may not hold at the individual level due to ecological fallacy [28]. Third, uncertainty in UN demographic projections may affect our BAPC forecasts, particularly in regions experiencing rapid political or economic transitions. Fourth, temporal changes in diagnostic practices (increased ultrasonography availability, updated clinical guidelines) may introduce period effects that confound true epidemiological trends [29]. Fifth, Our analysis is inherently limited by the ecological nature of GBD data, which precludes subgroup analyses by specific comorbidities (diabetes, hypertension, cardiovascular disease). However, population-level evidence suggests >70% of NAFLD patients ≥45 years exhibit metabolic multimorbidity, with diabetes prevalence ranging from 15–25% in medium-SDI regions to 8–12% in high-SDI areas. Future studies utilizing individual patient data are needed to elucidate comorbidity-specific risk patterns and therapeutic responses in aging NAFLD populations. [30].

Future efforts should focus on standardizing data collection and diagnostic criteria, particularly in high-burden aging populations. Incorporating real-world clinical and metabolic biomarkers will enable more accurate projections and intervention strategies. Addressing these challenges is essential for formulating effective public health strategies against NAFLD in older adults. We should focus on future research that include cost-effectiveness of population-level screening in high-prevalence regions, validation FLI against transient elastography in resource-limited settings, novel biomarkers for fibrosis risk stratification in aging populations [31], and interventions addressing social determinants of NAFLD disparities [32].

## Conclusion

This study evaluated the global burden of non-alcoholic fatty liver disease (NAFLD) in adults who are 45 years old and older, covering the years from 1990 to 2021, revealing significant epidemiological shifts. The rates of incidence and prevalence, adjusted for age, rose by 18.3% and 24.5%, severally, with disproportionate burdens in middle-to-high SDI regions due to metabolic risks and aging demographics. While women showed higher incidence rates, men exhibited consistently elevated mortality rates, highlighting unmet intervention needs. Projections to 2035 suggest rising incidence (particularly among women) alongside modest declines in mortality rates and life years adjusted for disability, emphasizing the need for prevention strategies designed for particular age groups and genders.

## Author contributions

**Conceptualization:** Chenyang Wang.

**Data curation:** Qian Wang, Jieru Guo, Shuang Liu.

**Formal analysis:** Qian Wang, Jieru Guo, Shuang Liu.

**Funding acquisition:** Chenyang Wang.

**Investigation:** Xuebin Cao, Zhirong Guo.

**Methodology:** Xuebin Cao.

**Project administration:** Zhirong Guo.

**Resources:** Chenyang Wang.

**Visualization:** Shuang Liu, Long Rui, Liu Zheng.

**Writing – original draft:** Qian Wang.

**Writing – review & editing:** Chenyang Wang.

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
