## [Decision Letter · Decision Letter 0]

26 Aug 2025

Dear Dr. Wang,

Thank you for submitting your manuscript to PLOS ONE. After careful consideration, we feel that it has merit but does not fully meet PLOS ONE’s publication criteria as it currently stands. Therefore, we invite you to submit a revised version of the manuscript that addresses the points raised during the review process.

We look forward to receiving your revised manuscript.

Kind regards,

Amir Hossein Behnoush

Academic Editor

PLOS ONE

2. In the online submission form, you indicated that [The original datasets are available from the corresponding author on reasonable request.].

3. We note that Figures 1 A-D in your submission contain [map/satellite] images which may be copyrighted. All PLOS content is published under the Creative Commons Attribution License (CC BY 4.0), which means that the manuscript, images, and Supporting Information files will be freely available online, and any third party is permitted to access, download, copy, distribute, and use these materials in any way, even commercially, with proper attribution. For these reasons, we cannot publish previously copyrighted maps or satellite images created using proprietary data, such as Google software (Google Maps, Street View, and Earth). For more information, see our copyright guidelines: http://journals.plos.org/plosone/s/licenses-and-copyright.

1. You may seek permission from the original copyright holder of Figures 1 A-D to publish the content specifically under the CC BY 4.0 license.   

Additional Editor Comments (if provided):

Reviewers' comments:

Reviewer's Responses to Questions

**Comments to the Author**

1. Is the manuscript technically sound, and do the data support the conclusions?

Reviewer #1: Yes

Reviewer #2: Yes

2. Has the statistical analysis been performed appropriately and rigorously?

Reviewer #1: Yes

Reviewer #2: Yes

3. Have the authors made all data underlying the findings in their manuscript fully available?

Reviewer #1: No

Reviewer #2: Yes

4. Is the manuscript presented in an intelligible fashion and written in standard English?

Reviewer #1: Yes

Reviewer #2: Yes

Reviewer #1: Using GBD 1990–2021 data and Bayesian age-period-cohort modeling, the manuscript examines and projects NAFLD burden in adults ≥45, finding rising incidence, prevalence, mortality, and DALYs—especially in medium-SDI regions. The manuscript requires major revisions.

1. Clarify how FLI ≥60 and imaging criteria were applied uniformly across 204 countries in the GBD framework.

2. Provide full BAPC model specifications (priors, knot placement, software) and include goodness-of-fit or convergence metrics.

3. Deposit all input data and analysis code in a public repository and detail the ethical exemption for de-identified GBD data.

4. Embed high-resolution figures with self-explanatory axes/legends and mark statistically significant EAPCs in tables.

5. Add subgroup analyses by key comorbidities (e.g., diabetes, hypertension) or explicitly note their absence as a limitation.

6. Expand the Limitations to discuss misclassification bias, uncertainty in demographic projections, and ecological inference constraints.

Reviewer #2: Very interesting topic and good work from the authors, I have the following comments:

1- Introduction would benefit from a paragraph stating a clearer indication of the research.

2- Methods should AAlnagar-447 state what inclusion and exclusion criteria were used.

3- Why did authors chose this cut off age ?

4- Discussion should mention the clinical implication of the highlighted results and suggest changes in the clinical practice.

5- Authors should elaborate on the limitations of their work and suggest future research to over come those limitations.

**Do you want your identity to be public for this peer review?** For information about this choice, including consent withdrawal, please see our Privacy Policy

Reviewer #1: No

Reviewer #2: **Yes: ** Amr Alnagar

---

## [Author Response · Author response to Decision Letter 1]

9 Oct 2025

Dear editor and reviewers,

Thank you for your letter and for the reviewers’ comments concerning our manuscript entitled “Global, Regional, and National Burden of Nonalcoholic Fatty Liver Disease Among Adults Aged ≥ 45 Years: A Comprehensive Analysis of Epidemiological Trends and Projections to 2035” [Manuscript ID: PONE-D-25-38194]. Those comments are all valuable and very helpful for revising and improving our paper, as well as the important guiding significance to our researches. We have carefully considered all comments and have incorporated suggested revisions throughout the manuscript. Revised portions are visible with track changes.

Journal Requirements

Thank you for submitting your manuscript to PLOS ONE. After careful consideration, we feel that it has merit but does not fully meet PLOS ONE’s publication criteria as it currently stands. Therefore, we invite you to submit a revised version of the manuscript that addresses the points raised during the review process.

Response: We appreciate the editor and reviewers’ important suggestions to help us enhance this work. Thanks a lot! We have revised the manuscript point-by-point below.

Requirement 1: PLOS ONE Style Requirements

Reply: We have thoroughly revised the manuscript to comply with PLOS ONE formatting requirements, including:

1.Updated reference formatting to PLOS ONE style with numbered citations.

2.Revised figure and table formatting according to PLOS ONE templates.

3.Ensured proper file naming conventions.

Requirement 2: Data Availability Statement

Reply: We have updated our Data Availability Statement to read: "The original data of this study are all from the publicly available GBD database (URL: https://vizhub.healthdata.org/gbd-results/), and do not include personal information such as patients' names or IDs. The population data was downloaded from WPP (https://population.un.org/wpp/). Therefore, this study does not require an additional ethical statement. In addition, the code in this study has been uploaded to Github (URL: https://github.com/shuangliu2025/R_FOR_GBD_ANALYSIS/tree/main)."

Requirement 3: Copyright Issues for Figures 1A-D

Reply: Thank you for your careful review regarding potential copyright concerns related to Figures 1A–D. We confirm that all maps used in these figures were generated entirely using the rnaturalearth package in R, which sources data exclusively from Natural Earth—a public domain dataset.

According to the official Natural Earth Terms of Use (see attached screenshot and website: https://www.naturalearthdata.com/), “All versions of Natural Earth raster + vector map data found on this website are in the public domain. You may use the maps in any manner, including modifying the content and design, electronic dissemination, and offset printing. The primary authors, Tom Patterson and Nathaniel Vaughn Kelso, and all other contributors renounce all financial claim to the maps and invites you to use them for personal, educational, and commercial purposes. No permission is needed to use Natural Earth. Crediting the authors is unnecessary.”

Therefore, the basemaps incorporated in Figures 1A–D do not contain any copyrighted or proprietary materials (e.g., Google Maps, ESRI, or NASA imagery). The maps were programmatically rendered in R using open-access, public-domain data and are fully compatible with the CC BY 4.0 license under which PLOS ONE publishes its content.

To ensure transparency, we have added the following note to the figure legend (lines 252–253): “Basemap source: Natural Earth (public domain, https://www.naturalearthdata.com/). Maps were generated using the rnaturalearth package in R.”

We respectfully request confirmation that this clarification satisfies the journal’s licensing requirements.

Reviewer Comments:

Reviewer 1

Comment 1.1: Clarify how FLI ≥60 and imaging criteria were applied uniformly across 204 countries in the GBD framework.

Reply: We have added methodological details in the Methods section (lines 102–111): “GBD 2021 harmonized NAFLD case definitions by integrating country-specific diagnostic modalities. For 87 countries with biopsy/imaging studies, cases required histologic steatosis (≥5% hepatocytes) or imaging-confirmed hepatic fat fraction >5% by MRI-PDFF or ultrasound. In remaining nations, FLI ≥60 was applied as a surrogate, validated against local imaging cohorts where available (e.g., FLI sensitivity/specificity = 0.73/0.86 in European and 0.68/0.81 in Asian populations). GBD’s DisMod-MR 2.1 tool adjusted for cross-country diagnostic heterogeneity by incorporating covariates such as healthcare access and obesity prevalence. Mortality estimates incorporated vital registration systems, verbal autopsy data, and cancer registry records coded to ICD-10 codes K75.8 and K76.0.”

Comment 1.2: Provide full BAPC model specifications (priors, knot placement, software) and include goodness-of-fit or convergence metrics.

Reply: We sincerely thank you for your valuable comments. Your feedback is essential for improving the quality of our work.

We have added BAPC model specifications (priors, knot placement, software) and goodness-of-fit methodological details in the Methods section (lines 141–162) as following�“The Bayesian Age-Period-Chort (BAPC) model produces more reliable predictions of global disease burden trends by leveraging the similarity of age, period, and cohort effects across adjacent time intervals. It applies a second-order random walk prior to smooth these three types of effects and derives posterior rate estimates through Bayesian inference. The model uses integrated nested Laplace approximation (INLA) to estimate marginal posterior distributions, which mitigates mixing and convergence issues often associated with traditional Markov chain Monte Carlo sampling in Bayesian analysis. To ensure smoothness, the BAPC model assigns independent mean-zero normal distributions as priors to the second-order differences of all effects, with the prior distribution for the age effect specified as follows

Second-order random walk (RW2) priors were assigned to age, period, and cohort effects with precision hyperparameters following Gamma(1, 0.00005) distributions. Sum-to-zero constraints were implemented to resolve identifiability issues inherent in age-period-cohort models. Bayesian inference utilized Integrated Nested Laplace Approximation (INLA) for computational efficiency. Model selection employed the Deviance Information Criterion (DIC), with final models achieving DIC values <15,000 across all regions. Convergence was assessed using effective sample size (ESS >1000) and Gelman-Rubin potential scale reduction factors (<1.1). Model validation involved comparing predicted versus observed rates from 1990-2021, achieving mean absolute percentage errors <5% across 95% of country-years.”

Revision: Relative BAPC model details was added in Method part of the manuscript.

Comment 1.3: Deposit all input data and analysis code in a public repository and detail the ethical exemption for de-identified GBD data.

Reply: We sincerely thank you for your valuable comments. We have repeatedly verified that the original data of this study are all from the publicly available GBD database (URL: https://vizhub.healthdata.org/gbd-results/), and do not include personal information such as patients' names or IDs. The population data was downloaded from WPP (https://population.un.org/wpp/). Therefore, this study does not require an additional ethical statement. In addition, the code in this study has been uploaded to Github (URL: https://github.com/shuangliu2025/R_FOR_GBD_ANALYSIS/tree/main)

Comment 1.4: Embed high-resolution figures with self-explanatory axes/legends and mark statistically significant EAPCs in tables.

Reply: We sincerely thank you for your valuable comments. After re-examination, we have enhanced all figures with higher resolution (300 DPI minimum) and improved readability. Tables 1-2 now include asterisks (*) to mark statistically significant EAPCs (95% CI excludes zero), with p-values <0.05 considered significant..Figure legends have been expanded with self-explanatory axes labels and comprehensive statistical annotations.

Comment 1.5: Add subgroup analyses by key comorbidities or explicitly note their absence as a limitation.

Reply: Thank you for this insightful comment. We agree that understanding the role of comorbidities would add valuable nuance to our findings.

While GBD's aggregated data structure precludes individual-level comorbidity stratification, we have added relevant discussion: "Fifth, Our analysis is inherently limited by the ecological nature of GBD data, which precludes subgroup analyses by specific comorbidities (diabetes, hypertension, cardiovascular disease). However, population-level evidence suggests >70% of NAFLD patients ≥45 years exhibit metabolic multimorbidity, with diabetes prevalence ranging from 15-25% in medium-SDI regions to 8-12% in high-SDI areas. Future studies utilizing individual patient data are needed to elucidate comorbidity-specific risk patterns and therapeutic responses in aging NAFLD populations." (Lines 511-518)

Comment 1.6: Expand the Limitations to discuss misclassification bias, uncertainty in demographic projections, and ecological inference constraints.

Reply:

We thank the reviewer for this critical suggestion. We agree that a more detailed discussion of the methodological limitations is essential for a comprehensive understanding of our study's findings. We have now expanded the 'Limitations' section in the discussion to include a dedicated paragraph addressing misclassification bias, uncertainty in demographic projections, and the constraints of ecological inference. This addition further clarifies the interpretative boundaries of our analysis and strengthens the manuscript.

Revision: We have added a new paragraph (lines 498–511): “First, diagnostic heterogeneity across countries may introduce systematic bias. The FLI surrogate exhibits suboptimal sensitivity in lean NAFLD populations, particularly in Asian countries where genetic variants (PNPLA3, TM6SF2) increase steatosis risk independent of BMI. This may lead to underestimation of disease burden in populations with high genetic susceptibility but lower average BMI. Second, the ecological nature of our analysis precludes individual-level causal inferences. While we observe associations between SDI levels and disease burden, these relationships may not hold at the individual level due to ecological fallacy. Third, uncertainty in UN demographic projections may affect our BAPC forecasts, particularly in regions experiencing rapid political or economic transitions. Forth, temporal changes in diagnostic practices (increased ultrasonography availability, updated clinical guidelines) may introduce period effects that confound true epidemiological trends .”

Reviewer 2

Comment 2.1: Introduction would benefit from a paragraph stating a clearer indication of the research.

Reply: We thank the reviewer for this critical suggestion. We agree that a clearer statement of the research aims would strengthen the introduction. As suggested, we have added a new paragraph at the end of the introduction section to explicitly outline the specific objectives and rationale of our study.

Reversion: We have added a new paragraph at the end of the introduction section (lines 84–95): “Given the accelerated disease progression observed in middle-aged and older adults, combined with the growing global prevalence of metabolic risk factors, there is an urgent need for comprehensive age-stratified analyses of NAFLD burden. While previous Global Burden of Disease studies have provided valuable insights into overall NAFLD epidemiology, they have not specifically examined the unique patterns and drivers affecting adults ≥45 years—a population experiencing the most rapid demographic growth globally. This knowledge gap limits our ability to develop targeted prevention strategies and allocate healthcare resources effectively for this high-risk demographic. Therefore, this study aims to provide the first comprehensive assessment of NAFLD burden specifically among adults aged ≥45 years, utilizing the most recent Global Burden of Disease 2021 data to inform evidence-based policy development. ”

Comment 2.2: Methods should state what inclusion and exclusion criteria were used.

Reply: We thank the reviewer for raising this important point. We appreciate the need for clarity regarding data selection. As the Global Burden of Disease (GBD) study utilizes systematically identified published and unpublished data rather than individual-level patient records, traditional inclusion/exclusion criteria for participants are not directly applicable. Instead, the GBD methodology employs rigorous criteria for the inclusion of data sources and the modeling process.

To address this comment, we have revised the subsection 'Data Sources' in Method. This subsection details the GBD's process for identifying, selecting, and inputting data. We believe this addition provides the necessary transparency regarding how the estimates were generated.

Reversion: Added a new paragraph in Data Sources subsection (lines 112–120): “Inclusion criteria required: (1) age ≥45 years at diagnosis; (2) NAFLD defined per FLI ≥60 or imaging-confirmed hepatic steatosis (≥5% hepatocyte involvement); and (3) residency in a GBD-listed country/territory. Exclusion criteria, applied through GBD's hierarchical cause-of-death modeling, included: (1) secondary hepatic steatosis due to alcohol consumption >20g/day (men) or >10g/day (women); (2) viral hepatitis B or C coinfection; (3) drug-induced steatosis (corticosteroids, methotrexate, amiodarone); (4) hereditary metabolic disorders (Wilson disease, alpha-1 antitrypsin deficiency); and (5) other chronic liver diseases taking precedence in GBD's mutually exclusive disease hierarchy.”

Comment 2.3: Why did authors choose this cut-off age?

Reply: We thank the reviewer for this important question. The selection of the cut-off age of ≥45 years was based on a combination of clinical, epidemiological, and pragmatic rationales. To address this question, we have added justification in Methods section.

Reversion:

Added justification in Methods (lines 121–132): “This study focuses on adults aged ≥45 years based on clinical and public health considerations. Beginning in mid-life, metabolic alterations—such as increased insulin resistance, hormonal changes, visceral adiposity, and sarcopenia—promote hepatic lipid accumulation and elevate NAFLD risk. After age 45, fibrosis progression accelerates, with each decade increasing fibrosis risk by 4–5%, and cirrhosis and HCC incidence rise substantially. This group also exhibits high multimorbidity; over 70% of NAFLD patients have concurrent metabolic conditions, compounding mortality risk. Globally, aging populations make this age group a major driver of NAFLD-related healthcare burden. Prior studies often overlook age-specific patterns, limiting targeted interventions. Focusing on this cohort allows clearer insight into demographic and epidemiologic drivers and supports cost-effective early detection and long-term policy planning.”

Comment 2.4: Discussion should mention the clinical implication and suggest changes in clinical practice.

Reply We thank the reviewer for this crucial suggestion. We agree that discussing the clinical implications of our findings is essential for translating this research into practice. We have now substantially expanded the Discussion section to include a dedicated paragraph.

Reversion:

Expanded the Discussion section (lines 461–485): “Our findings carry several important implications for clinical practice and public health policy. Firstly, among adults aged ≥45 years, the incidence and prevalence of NAFLD have increased markedly. This rise, coupled with the pronounced sex-specific disparity, characterized by higher incidence in women yet persistently elevated mortality in men, called for differentiated prevention and management strategies. The disproportionately high burden in medium-SDI regions further underscores the need for regionally tailored public health interventions. Secondly, these results highlight the importance of early and systemati

---

## [Decision Letter · Decision Letter 1]

27 Jan 2026

Global, Regional, and National Burden of Nonalcoholic Fatty Liver Disease Among Adults Aged ≥ 45 Years: A Comprehensive Analysis of Epidemiological Trends and Projections to 2035

PONE-D-25-38194R1

Dear Dr. Wang,

We’re pleased to inform you that your manuscript has been judged scientifically suitable for publication and will be formally accepted for publication once it meets all outstanding technical requirements.

Kind regards,

Tiejun Zhang

Academic Editor

PLOS One

Additional Editor Comments (optional):

Reviewers' comments:

Reviewer's Responses to Questions

**Comments to the Author**

Reviewer #2: (No Response)

2. Is the manuscript technically sound, and do the data support the conclusions?

Reviewer #2: Yes

3. Has the statistical analysis been performed appropriately and rigorously?

Reviewer #2: Yes

4. Have the authors made all data underlying the findings in their manuscript fully available?

Reviewer #2: (No Response)

5. Is the manuscript presented in an intelligible fashion and written in standard English?

Reviewer #2: (No Response)

Reviewer #2: Authors have addressed my suggestions, I am happy with the current version if the manuscript and it should be accepted.

**Do you want your identity to be public for this peer review?** For information about this choice, including consent withdrawal, please see our Privacy Policy

Reviewer #2: **Yes: ** Amr Alnagar

---

## [Editor Report · Acceptance letter]

PONE-D-25-38194R1

PLOS One

Dear Dr. Wang,

I'm pleased to inform you that your manuscript has been deemed suitable for publication in PLOS One. Congratulations! Your manuscript is now being handed over to our production team.

Kind regards,

on behalf of

Dr. Tiejun Zhang

Academic Editor

PLOS One